# Analysis and Comparison of GPS Precipitable Water Estimates between Two Nearby Stations on Tahiti Island

**DOI:** 10.3390/s19245578

**Published:** 2019-12-17

**Authors:** Fangzhao Zhang, Jean-Pierre Barriot, Guochang Xu, Marania Hopuare

**Affiliations:** 1Geodesy Observatory of Tahiti, University of French Polynesia, Faa’a 98702, French Polynesia; jean-pierre.barriot@upf.pf (J.-P.B.);; 2Institute of Space Sciences, Shandong University, Weihai 264209, China

**Keywords:** precipitable water, GNSS, zenith total delay, weighted mean temperature, insolation, altitude, wind

## Abstract

Since Bevis first proposed Global Positioning System (GPS) meteorology in 1992, the precipitable water (PW) estimates retrieved from Global Navigation Satellite System (GNSS) networks with high accuracy have been widely used in many meteorological applications. The proper estimation of GNSS PW can be affected by the GNSS processing strategy as well as the local geographical properties of GNSS sites. To better understand the impact of these factors, we compare PW estimates from two nearby permanent GPS stations (THTI and FAA1) in the tropical Tahiti Island, a basalt shield volcano located in the South Pacific, with a mean slope of 8% and a diameter of 30 km. The altitude difference between the two stations is 86.14 m, and their horizontal distance difference is 2.56 km. In this paper, Bernese GNSS Software Version 5.2 with precise point positioning (PPP) and Vienna mapping function 1 (VMF1) was applied to estimate the zenith tropospheric delay (ZTD), which was compared with the International GNSS Service (IGS) Final products. The meteorological parameters sourced from the European Center for Medium-Range Weather Forecasts (ECMWF) and the local weighted mean temperature ($T_{m}$) model were used to estimate the GPS PW for three years (May 2016 to April 2019). The results show that the differences of PW between two nearby GPS stations is nearly a constant with value 1.73 mm. In our case, this difference is mainly driven by insolation differences, the difference in altitude and the wind being only second factors.

## 1. Introduction

Atmospheric water vapor plays an essential role in atmospheric processes in terms of its intimate coupling to the greenhouse effect, the formation of clouds and rainfall, the atmospheric storm systems, the Earth’s meridional energy balance and atmospheric chemical reactions [1,2,3]. Precipitable water (PW) refers to the equivalent height or liquid water, expressed in millimeters (mm), of the total water vapor contained in an air column from the Earth’s surface to the top of the atmosphere [4]. Since Bevis et al. (1992) [5] first proposed Global Positioning System (GPS) meteorology to remote sensing atmospheric water vapor, GPS meteorology has become an important approach for water vapor detection with high accuracy, all-weather capability, high spatial temporal resolution, and low costs [6,7,8]. With the rapid development of the current global navigation satellite systems (GNSS) constellation, GNSS meteorology has the potential to provide precise PW values with high temporal resolution and is indeed widely used in many meteorological applications, such as numerical weather prediction, water vapor tomography, nowcasting, and severe weather event monitoring [9,10,11,12,13,14,15]. Moreover, water vapor not only contributes significantly to the opacity of the atmosphere in the infrared wavelengths as a greenhouse gas but is also the dominant source of phase fluctuations in radio sciences [16,17]. The PW values can be used to make first-order predictions of the opacity of the atmosphere, although there is little information regarding phase fluctuations. Butler (1998) made a rough local estimate of the PW values from the measurements of surface temperature and dew point by assuming that the water vapor is exponentially distributed in the atmosphere with respect to attitude above a given location [18].

Several international organizations, the most active being International GNSS Service (IGS) and European Center for Medium-Range Weather Forecasts (ECMWF) pool core GNSS stations around the world, essentially to provide data to maintain the International Terrestrial Reference frame (ITRF) and the Vienna mapping function 1 (VMF1) family of mapping functions (used to correct tropospheric delays). These stations are unevenly distributed around the world, with a repartition driven by the density of populations. Tahiti Island is an extreme case, with the THTI and FAA1 stations located 2.56 km apart and surrounded by nearly unlimited seas. The nearest IGS station is located on the Mangareva Island, at a distance of 1500 km from Papeete. As a result, these isolated stations have a heavy weight on worldwide networks, as they constrain geophysical models in a large radius. As we have the chance in Tahiti to have two nearly collocated but also isolated stations, it then makes sense to study the differences in their time series, in order to assess the errors that can be induced in global models by the data from such isolated stations.

Geomorphological and climate settings are described in Section 2. In Section 3, GPS settings of the THTI and FAA1 IGS stations are given. In Section 4, the estimation of GPS PW is presented in detail. The exponential distribution of PW with altitude is given in Section 5. The results and discussion are presented in Section 6, which includes the results of zenith total delay (ZTD), zenith non-hydrostatic (wet) delay (ZWD), PW, and their statistical analysis; the comparison between GPS PW and the PW values based on an exponential distribution with respect to attitude; and the correlation between ZWD, PW, and wind velocity, and the influence of local topography. Conclusions are presented in Section 7.

## 2. Geomorphological and Climate Setting

The two GPS stations of our study are located in the tropical Tahiti Island (see Figure 1), the main island of a swarm of 118 small islands in the South Pacific hemisphere. Tahiti is an extinct volcanic structure joining two shield volcanoes (Tahiti Nui and Tahiti Iti) separated by an isthmus. Tahiti Nui and Tahiti Iti are young in the geological sense. They were created on the ocean floor by a mantellic plume about 5 million years ago, and all volcanic activity ceased on Tahiti Nui about 30,000 years ago. After the peak of volcanic activity, the two sub-islands collapsed, creating a radial, highly dissected by erosion, network of valleys centered on the calderas. The mean slope of the volcanoes is around 8%, and the slopes of the flanks of the valleys can reach 40% to 70% [19] (Figure 1). The climate in Tahiti is very humid (meaning high PW values) [20] and characterized by dry (from May to October) and wet seasons (from November to April) that are governed by the South Pacific Convergence Zone (SPCZ) on the large scale and by the topography of a high volcanic island on the orographic scale [21]. The two GPS stations of this study are known in the IGS database as THTI and FAA1. Their altitude difference is 86.14 m, and their horizontal distance difference is only 2.56 km. Although they are close (Figure 1), their surroundings are quite different. FAA1, managed by Météo-France, is located at near sea-level, under the premises of the international airport, on a very large area that was flattened to host the airstrip. THTI is located on the main campus of University of French Polynesia, a little bit more inland, on a ridge crest with a slope of about 8%, at a distance from the sea of about 1.5 km. They belong to two distinct valleys, each one with its own micro-climate [22].

## 3. GPS Settings of the THTI and FAA1 IGS Stations

The THTI station consists of a SEPT POLARX5TR receiver with a LEIAR.R425.R3 antenna, and the FAA1 station consists of a SEPT POLARX4 receiver with a LEIAR.R4 antenna. GPS satellite precise orbits and clocks as well as consistent Earth-rotation parameters were provided by the Center for Orbit Determination in Europe (CODE). Together with the Bernese GNSS Software Version 5.2 with PPP approach, hydrostatic and wet Vienna Mapping Function 1 (VMF1) were used to estimate the THTI and FAA1 ZTD estimates with a resolution of one hour [23,24,25]. To validate the accuracy of our ZTD estimates, we made a comparison with the IGS Final troposphere products (ftp://cddis.gsfc.nasa.gov). Details of both data processing strategies (ours and IGS) are summarized in Table 1. The Bernese 5.2 instead of Bernese 5.0 is used by IGS products from the day of year 29 in 2017. The definitions of bias, root mean square (RMS) and standard deviation (STD)) are shown in Appendix A.

## 4. The Estimation of GPS PW Values

As one of the most important data in many meteorological departments, numerous efforts have been made to improve the accuracy of GPS PW values based on different GPS processing strategies. 

There are two common positioning strategies used by nearly all the GPS-derived PW retrieval systems. The first one is the precise point positioning (PPP) technique based on un-differenced approach, the second one is the relative positioning strategy based on the double-differenced approach [26,27,28,29]. Haase et al. (2003) [30] noted that both strategies provide similar PW accuracy when dealing with large sets of GNSS observations. Compared with relative positioning strategy, which needs to introduce long baselines (larger than 500 km) to weaken the spatial correlations between network stations, PPP strategy can operate on an isolated GPS receiver [31]. ZTD values, once all the other errors in GNSS observations have been modeled or corrected, such as phase center offset (PCO) and phase center variation (PCV) models [32], can be estimated precisely based on PPP technique with the high accuracy of GNSS satellite ephemeris and clock products [33]. The ZTD estimate can be divided into two additive components: The zenith hydrostatic delay (ZHD) and the ZWD. The ZHD estimates depend on the atmospheric pressure and can be estimated locally from the surface pressure values recorded at the GPS receiver station with mm accuracy [34]. Thereafter, the ZWD estimate is computed by subtracting the ZHD estimate from the ZTD estimate. The weighted mean temperature of the atmospheric column ($T_{m}$) plays a very important key in the progress of retrieving PW from the ZWD, which varies in space and in time. The weighted temperature $T_{m}$ can be estimated by using a linear function (y=ax+b) of the surface temperature $T_{s}$ [35,36]. Bevis et al. (1992) [5] noted that to get the best results, the constants *a* and *b* should be ”tuned” to specific areas and seasons. After the seminal work of Bevis et al., many additional efforts have been made to retrieve better GNSS PW estimates by developing better relationships, often based on regional and seasonal fits, between $T_{m}$ and $T_{s}$ (Bevis et al. (1994) [35], Ross and Rosenfeld (1997) [37], Mendes et al. (2000) [38], Wang et al. (2005) [36], Suresh Raju et al. (2007) [39], Yao et al. (2012) [40], Huang et al. (2018) [41] and Zhang et al. (2018) [20]).

The ZHD accounts for almost 90% of the total ZTD, and the ZWD accounts for about 10% [42]. ZHD can be calculated by adopting the Saastamoinen model, in mm, as given by [34]:(1)$$ZHD=2.2768\;P_{s}/f\left(\lambda ,\;H\right),$$
where f(λ,H)=1−0.00266·cos(2λ)−0.00028·H, λ(rad) is the station latitude, H(km) is the height of the surface above the ellipsoid, and $P_{s}\left(hPa\right)$ is the surface pressure. The ZWD is obtained, by definition, by subtracting the ZHD from the ZTD obtained from our GPS data processing, as shown:(2)ZWD=ZTD−ZHD,

The PW estimate is directly related to ZTD by Equation (3) [42]:(3)PW=Π·ZWD,
where the conversion parameter Π varies as a function of the weighted mean temperature of the atmosphere $T_{m}$ (in Kelvin) [36]:(4)$$\Pi =\frac{10^{6}}{\rho \cdot R_{v}\left(\frac{k_{3}}{T_{m}}+k_{2}^{\prime }\right)},$$
where $k_{3}$ and $k_{2}^{\prime }$ are atmospheric refractivity constants [35]:(5)$$k_{2}^{\prime }=22.1\;K/\mathrm{mb},$$
and
(6)$$k_{3}=3.739\cdot 10^{5}{\;K}^{2}/\mathrm{mb},$$
where $\rho =1000\;kg/m^{3}$ is the density of liquid water, $R_{v}=461.495\;J/kg\cdot K$ is the specific gas constant for water vapor [42,43]. To calculate $T_{m}$, vertical profiles of water vapor and temperature are needed [42]. Many scientists have explored different kinds of linear relationship between $T_{m}$ and $T_{s}$ from an a priori knowledge of the water vapor contents of the atmosphere from radio soundings, based on global and regional [35,39,44,45,46,47]. Ross and Rosenfeld [37] noted that the relationship between $T_{m}$ and $T_{s}$ changes with geographic altitude as well as season by using 53 global radiosonde stations. In this work, we used specific $T_{m}$ models (see Table 2), that we published in a previous paper (Zhang et al. [20]), calibrated against three years of GPS ZWD and raw radiosonde (RS) data on Tahiti Island.

## 5. The Exponential Distribution of PW with Altitude

A rough estimate of PW can be gained from the measurements of surface temperature and dew point by assuming that the water vapor is exponentially distributed in the atmosphere above a given location [16]. If we consider a column of liquid water of height *h* with cross-sectional area *A*, the mass of PW in the atmosphere can be written as [18]:
(7)$$M_{h}=\rho \cdot A\cdot h,$$
where ρ is the density of liquid water. Meanwhile, the mass of the water vapor in an atmospheric column with cross-sectional area *A* is:
(8)$$M_{wv}=A\cdot m_{w}\cdot \int_{Z_{s}}^{\infty }n_{wv}\;dZ,$$
where Z is the altitude, $m_{w}$ is the mass of a molecule of water (18 amu). $n_{wv}$ is the density number of water vapor molecules, and we consider that the water vapor is distributed exponentially (similar to the bulk of the lower atmosphere) with an exponential decaying distribution as a function of Z like [18]:
(9)$$n_{wv}=n_{s}\cdot e^{-\left(Z-Z_{s}\right)/H},$$
where *H* is a scale height of water vapor. $n_{s}$ is the number of water vapor molecules at the surface ($Z_{s}$), which is related to the water vapor partial pressure ($P_{w}$) and the surface temperature $T_{s}$ in Kelvin by [17]:
(10)$$n_{s}=\frac{P_{w}}{K\cdot T_{s}},$$
and
(11)$$P_{w}=e^{1.81}e^{a\frac{T_{d}\left({}^{{}^{\circ}}C\right)}{T_{d}\left({}^{{}^{\circ}}C\right)+237.7}}=6.11\frac{RH}{100}e^{17.271\frac{T_{s}\left({}^{{}^{\circ}}C\right)}{237.7+T_{s}\left({}^{{}^{\circ}}C\right)}},$$
and
(12)$$T_{d}\left({}^{{}^{\circ}}C\right)=\frac{b\gamma }{a-\gamma },$$
and
(13)$$\gamma =a\frac{T_{s}\left({}^{{}^{\circ}}C\right)}{b+T_{s}\left({}^{{}^{\circ}}C\right)}+\ln\left(RH/100\right),$$
where *K* is the Boltzmann constant and *RH* is the relative humidity, $T_{d}$ is the dew point temperature in degrees Celsius $\left({}^{{}^{\circ}}C\right)$, a = 17.271 and b = 237.7. $T_{s}\left({}^{{}^{\circ}}C\right)$ is the surface temperature in ${}^{{}^{\circ}}C$ [17].

Observationally, the scale height of water vapor (*H*) in the Earth’s atmosphere is between 1.5 and 2 km [18]. Butler (1998) [16] assumed the scale height of water vapor to be 1.5 km. Vicente and Pulido (2012) [17] investigated the annual average scale (1340 ± 120 m) from data obtained by two balloons, which provided data for pressure, height, temperature, dew point, relative humidity, and other parameters along their upwards trajectories through the atmosphere from 630 m to 30 km. 

In order to find the integrated exponential PW (PWE) from $Z_{s}$ to $Z_{0}$, we have to equate the mass of the water vapor to that of the liquid (Equations (7) and (8)). Taking into account that the difference in altitude is very small, meaning that we can equate $e^{-\left(Z_{0}-Z_{s}\right)/H}$ to $1-\left(Z_{0}-Z_{s}\right)/H$, we get:(14)$$PWE\;=\frac{m_{w}\cdot n_{s}}{\rho }\cdot \left(Z_{0}-Z_{s}\right),$$
This quantity is independent of *H* up to the second order. Therefore,
(15)$$PWE\left(FAA1-THTI\right)=\frac{m_{w}\cdot n_{s}}{\rho }\cdot \left(Z_{sTHTI}-Z_{sFAA1}\right),$$

Equation (15) shows the difference of PW values at THTI and FAA1 as a function of their difference in altitude. $n_{s}$ can be relative to either THTI or FAA1, the corresponding PWE values will be very close.

## 6. Results and Discussion

### 6.1. ZTD Estimates for THTI and FAA1

In this section, we give the comparisons of THTI ZTD and FAA1 ZTD estimates based on our GPS data processing over three years (May 2016 to April 2019), with respect to the IGS Final tropospheric products. The temporal resolution is one hour. As displayed in Figure 2a,c, it can be noticed that the GPS-derived ZTD from our GPS data processing agrees very well with the corresponding ZTD from IGS Final products, for both stations (except from 9 July to 13 August 2018 with no GPS data measurements). Figure 2b,d and Table 3 show the distributions of their differences, after removing off the outliers (±3 sigma). They are close to Gaussian distributions. The ZTD differences are probably mainly due to the details of the algorithms and strategy used, with no significant systematic biases. The root mean square (RMS) values of ZTD differences are 5.37 mm at the THTI site and 8.21 mm at the FAA1 site. FAA1 is located near sea-level, and this may be the cause of its slightly lower accuracy with respect to the IGS Final products.

Figure 3 shows the inter-comparisons of our ZTD estimates (Figure 3a) and IGS ZTD estimates (Figure 3c) between THTI and FAA1 at the same epoch, and Table 4 summarizes the results. It can be noticed that there is a larger bias (around 30 mm) between our THTI ZTD and FAA1 ZTD estimates, also seen between the IGS THTI ZTD and IGS FAA1 ZTD estimates. The difference in altitude between the two GPS stations may play a role here, with different meteorological parameters entering the mapping functions used in the respective processing. It must be emphasized that the VMF1 mapping functions (Table 1) used in the Bernese 5.2 are inputting ECMWF meteorological data (see following Section 6.2).

### 6.2. Meteorological Data

The site-specific pressure $P_{s}$ and temperature $T_{s}$ are very important parameters to derive the precise GPS PW estimates (see Equations (2) to (4)). In this study, these meteorological parameters were sourced from the gridded (2.0° × 2.5° latitude-longitude) VMF1 (vmf1_g) data of the ECMWF (http://ggosatm.hg.tuwien.ac.at) with a 6 hour temporal resolution, including hydrostatic/wet coefficient, hydrostatic/wet zenith delay, mean temperature, pressure/temperature at the site, water vapor pressure, and orthometric height of the station (using geoid EGM96).

In order to assess the reliability of the ECMWF pressure and temperature data, we compared firstly their values with the local values recorded at the THTI station for the whole year of 2018. The THTI recorded data are from the co-located rain gauge station (a few meters distance from the GPS antenna), with a 1 min sampling, and include temperature, relative humidity, pressure, rainfall accumulation, and rainfall intensity [48]. Figure 4 shows the variations of the pressure and temperature from ECMWF (with a 1 hour linear sub-interpolation) and the local rain gauge station at THTI. It can be seen that the two meteorological datasets are consistent. Table 5 summarizes the comparison results—the bias of pressure and temperature is 0.58 hPa and 0.26 K, respectively, and the associated RMS is 0.87 hPa and 2.21 K. 

Then, the ZHD is calculated according to Saastamoinen model (Equation (1)) based on the local pressure and the ECMWF pressure. Figure 5 shows the different ZHD values at the THTI station during the whole year of 2018. The ZHDs derived from local pressure and those from the ECMWF are very close. Table 5 summarizes the statistical results between the ZHDs from the observed pressure values and those from the ECMWF. The ZHDs as derived from the ECMWF data are in error by about 1.32 mm, and the RMS is about 1.99 mm compared to the local pressure values. Therefore, the ECMWF meteorological data meet the requirement of the level of accuracy [23] and can be used for our following data processing.

Figure 5a,c shows the time series of pressure and temperature for the THTI and FAA1 sites, as extracted from the ECMWF database. The temporal resolution is six hours. Their differences are shown in Figure 5b,d and summarized in Table 6. The standard pressure difference between THTI and FAA1 coming from the difference in altitude should be −10.30 hPa (green dots in Figure 5b), instead of the −9.81 hPa (Table 6) value from the ECMWF database, according to the standard formula $P=1013.25\cdot {\left(1-0.0000226\cdot H\right)}^{5.225}$ [23,49]. This is coherent with the bias in pressure between local THTI and ECMWF THTI data (0.58 hPa) (Table 5). The difference in temperature between THTI and FAA1 should be −0.56 K (green dots in Figure 5d) according to the standard lapse rate of 0.0065 K/m ([50,51] with a sea level temperature of 288 K). This is also coherent with the bias in temperature between local THTI and ECMWF THTI data (0.26 K) (Table 5). 

### 6.3. ZWD Estimates for THTI and FAA1

The ECMWF pressure with a 1 hour linear sub-interpolation is used as our meteorological data to estimate the ZHDs based on the Saastamoinen model (Equation (1)) for three years (May 2016 to April 2019). The ZWD estimates were obtained by subtracting the corresponding ZHDs from the ZTDs obtained from our GPS data processing (see Equation (2)). The variations of THTI and FAA1 ZWDs are shown in Figure 6a. The bias of their differences is −8.12 mm, and the RMS is 10.76 mm (Table 7).

### 6.4. PW Estimates for THTI and FAA1

In this section, the GPS PW estimates were derived based on the ECMWF meteorological data and the local $T_{m}$ models (Table 2) for three years (May 2016 to April 2019). Figure 7a,c shows the time series of PW based on an all-season $T_{m}$ model and seasonal (wet and dry) $T_{m}$ model for THTI and FAA1 stations are similar. Figure 7b,d shows the variations of PW differences based on different $T_{m}$ models for THTI and FAA1 stations in the dry and wet seasons. Table 8 summarizes the statistical results. There is a larger bias and a smaller STD induced by the choice of $T_{m}$ model in the dry season than in the wet season at the two stations. 

Figure 8 shows the PW differences based on different $T_{m}$ models between two nearby GPS stations (THTI and FAA1). It can be seen that some large PW values (>70 mm) are showing up in the wet season due to the very humid climate of Tahiti. The FAA1 PW estimates are systematically higher than THTI PW estimates. Based on an all seasons’ $T_{m}$ model, the bias of their PW differences is −1.73 mm, and based on seasonal $T_{m}$ model, the bias of their PW differences is −1.82 mm (Table 9). The same GPS strategy was used during our GPS data processing for THTI and FAA1. It is natural to think that the difference in altitude may play a role here.

### 6.5. Statistical Analysis

The question of focus is now: Are the time series of differences in ZWD and PW estimates between the two stations “signal” or “noise”? To assess this question, we first made cross-comparisons between ZWD and PW time-series estimates versus a normal distribution by using quantile–quantile (QQ) plots (Figure 9a,d) for the THTI and FAA1 stations. All the distributions of ZWD and PW estimates shown in the figures are clearly non-Gaussian, with heavy tails, meaning that extreme values are more likely to be found than in a normal distribution. This has been noted for PW estimates by Iassamen et al. (2009) [52] who proposed a Weibull distribution. This heavy tails distribution can also be seen in the distributions of the differences between the PW and ZWD values (Figure 9e,f), albeit one expects that these differences will be naturally closer to a “white noise”, i.e., a Gaussian distribution. Figure 9g–j shows that the time series of ZWD and PW differences are also statistically unrelated to the ZWD and PW native series.

Secondly, we did a Fourier analysis (spectra) of the ZWD (Figure 10a,b) and PW (Figure 10c,d) time series of THTI and FAA1 and their differences and a comparison with the expected white noise spectrum. The large peak on the right of each figure corresponds to the annual cycle. A daily cycle can also be seen. It is important to note that the power spectrum of the time series of the differences lies over the white noise spectrum curves from 0 to 180 cycles/year, with spikes for 1 cycle/year (annual signature), 365 cycles/year (diurnal signature) and harmonics. This clearly means that there is a deterministic signal in the differences between THTI and FAA1 for the ZWD and PW estimates.

### 6.6. Comparisons between GPS PW and PWE for THTI and FAA1

In this section, we analyze the differences of PW values between THTI and FAA1 based on exponentially derived PW estimates (PWE, Equation (15)) to see if the differences in the PW time series can be attributed to the difference in altitude. 

For the calculation of PWE (THTI) and PWE (FAA1), the dew point temperature or the relative humidity and the surface temperature are needed (Equation (11)). The FAA1 site has dew point temperature records, which can be retrieved from the National Oceanic and Atmospheric Administration (NOAA) Integrated Surface Database (ISD) (https://www.ncdc.noaa.gov/isd). ISD includes numerous parameters such as wind speed and direction, wind gust, temperature, dew point, cloud data, sea level pressure, altimeter setting, station pressure, and various other elements as observed by each station [53]. However, THTI site does not have collocated NOAA data records, and the relative humidity is retrieved from a collocated rain gauge station operating since 2018 (see Section 6.2) [48]. 

Therefore, in order to maintain consistency between the PWE estimates of FAA1 and THTI, we chose to use for both sites ECMWF meteorological data, as we did for the GPS processing of PW values. We validated this choice by a cross-comparison between all the available temperature records at the two sites. To complete this cross-comparison, we first analyzed, for the whole year of 2018, the ECMWF temperature values with respect to the NOAA values for the FAA1 station. The bias between NOAA recorded temperatures and the corresponding ECMWF values is 0.87 °C with an RMS of 2.34 °C (Figure 11a and Table 10). For the THTI station, the bias between the rain gauge recorded temperatures and the corresponding ECMWF values is 0.26 **K** with an RMS of 2.21 **K** (Figure 4 and Table 5). The bias between the FAA1 NOAA recorded temperature and the THTI rain gauge temperature is 1.23 °C with an RMS of 1.44 °C (Figure 11b and Table 10). These differences are of the same order as the differences between the ECMWF temperature records at the two sites (bias of −0.60 **K** with an RMS of 0.60 **K** (Figure 5 and Table 6). This demonstrates that all temperature records are coherent down to 1 °C accuracy, validating the use of ECMWF values for both sites for the computation of PWE estimates.

Figure 12a shows the variations of the difference of GPS-PWs and PWEs. The scatter for the GPS-PWs differences between the two stations is a lot more pronounced that the corresponding scatter for the PWE values, revealing a large daily variability. Table 11 summarizes the statistical results. It can be seen that the bias of the difference of GPS-PWs is 1.75 mm and close to the bias (about 1.50 mm) of the differences of PWEs. Figure 12b and Table 12 show that the linear relation (*y* = *ax* + *b*) between GPS-PW differences and PWE differences. If these differences were only driven by the difference in altitude with respect to an exponential decay of the water vapor, this relationship should be a line with a slope value of one (*y* = *x*) (the black line in Figure 12b). Nevertheless, the cloud of data points in Figure 12b has almost a round shape, indicating that the differences in PW values cannot be attributed to the difference in altitude if we assume this exponential decay. The difference in altitude certainly plays a role, but a minor one.

### 6.7. Correlation between ZWD, PW, and Wind Measurements

In this section, we analyze the differences of ZWD and PW values between THTI and FAA1 during the period of 2017–2018 (we had at our disposal two full seasonal cycles) to see if the differences in these time series can be attributed to an effect induced by the wind. We used two wind datasets, the first one relative to a 10 m altitude, from a meteorological mast collocated with the FAA1 GNSS receiver (ftp://ftp.ncdc.noaa.gov/pub/data/noaa) and the other one relative to a 1000 m altitude, from the RS balloon station of the airport, also collocated with FAA1.

Figure 13a,c shows the variations of the 10 days averaged values of the wind velocity from NOAA and RS with the 10 days averaged values of ZWD values (Figure 13a) and PW values (Figure 13c). The variations of the corresponding averaged ZWD and PW differences with the wind velocity are shown in Figure 13b,d. As displayed, the averaged wind velocity values from RS (1000 m) are larger than those from NOAA (10 m), as there is less friction with the surface.

Table 13 summarizes the statistical results of the correlation with the wind. The correlations of the PW differences with the wind at 1000 m are at a 40% level, showing that the wind certainly plays a role in PW variations. The value of the same correlation with the surface wind is at a 15% level, almost not significant from a statistical point of view. However, we looked in further detail at the surface wind flow, that is to say wind speed and direction. PW is a vertical integral value; therefore, moist air parcels converging on the island would cause PW increases, and moist air parcels diverging would cause PW decreases.

Land–sea contrast initiates sea breeze during the day and land breeze at night. The wind mast at 10 m is located on the north-western coast of Tahiti, hence on the leeward side. Therefore, the breeze circulation tends to develop easily and frequently in this shielded area.

Sea breeze circulation develops during the day due to the temperature and pressure difference between the island and the surrounding ocean. Moist air parcels tend to converge towards the island, and clouds form and agglomerate on the mountains in the middle of the day. At night the process reverses; the air parcels hurtle down the valleys towards the ocean. In this case, moist air parcels go away from the island, hence there is divergence of the flow. 

Wind direction can therefore give insight to determine whether there is convergence and ascent (i.e., PW increase) or not. A closer look at the wind mast location on the island and the wind roses plotted on Figure 14 clearly shows that wind is coming from north-east (NE), north-west (NW) and south-west (SW) more often during the day (08:00 to 17:00) and air parcels are converging towards the island (Figure 14a). However, east and south-east (SE) winds draw the air parcels away from the island during night time (20:00 to 05:00) (Figure 14b).

In Figure 15, we plotted histograms of wind direction by considering three wind speed intervals: 0 to 2 m/s, 2 to 4 m/s, and 4 to 10 m/s (hourly wind direction and speed from NOAA at FAA1 station). These histograms show interesting wind flow characteristics. Considering low wind speed, one can notice that the corresponding wind direction comes from north-east and south-east quadrants (Figure 15a). The east and south-east directions, corresponding to the land breeze circulation and hence divergence (because the air parcels go away from the island) tend to “disappear” for moderate wind speeds (Figure 15b). Wind with high speed, coming from north-east, north-west, and south-west directions converge on the island, therefore causing ascent and humidity rise (Figure 15c). These focusing and defocusing mechanisms are the causes of the weak correlations seen in Table 13.

### 6.8. Influence of the Topography on ZWD and PW Estimates

As displayed in Figure 1, although the THTI and FAA1 stations are only 2.56 km apart, they are located in completely different settings. The FAA1 is located in the large, artificially flattened area of the airport a few meters from the sea, while the THTI is located above a ridge with a slope of about 8% (Figure 1). This difference in the topography settings is mainly due to the conical shape of the volcano, meaning that their mean insolation is different. The contents of water vapor in the atmosphere are driven in part (around 10% to 15% [54]) by land–surface evapotranspiration. This effect is certainly larger in the tropical rain forest of Tahiti [55]. Land evapotranspiration (soil evaporation plus plant transpiration) is driven, for a large part, by insolation ([56,57] and Figure 16), the vegetation cover [58], and the surface wind and surface temperature [59,60]. Research about evapotranspiration is a very active field nowadays. Paltineanu et al. (2012) [61] discussed evapotranspiration versus climatic water deficit. Falamarzi et al. (2014) [62] estimated evapotranspiration from temperature and surface wind speed data using artificial and wavelet neural networks (WNNs).

At the latitude of Tahiti, maximum mean insolation essentially corresponds to horizontal terrains. In other words, mean insolation will be lower on the slopes, leading to less evapotranspiration and then a little less PW in the atmosphere. This can be seen in another way—when you are on a slope, part of the uphill sky is masked, leading to less mean insolation. To give precise numbers is difficult, as the relation between PW and evapotranspiration must be certainly averaged over some area. As the topography of the volcano is very rough, with extremely large small-scale variations, this averaging is almost impossible. For example, THTI is on a crest ridge with an 8% slope, but with nearly a 50 m cliff on one side (see Figure 1b). This is also complicated by the changes in vegetation cover with the altitude that can reach up to 2200 m in Tahiti [63]. 

This influence of insolation can be partially tested by doing the Fourier analysis of the ratio (PW (THTI)−PW (FAA1))/PW (FAA1). As the two stations are very close, the insolation that acts as a multiplicative factor in the determination PW values through evapotranspiration is nearly the same at the top of the atmosphere for both stations. By making this ratio, which has a mean value of −0.0355 ± 0.0002 (Figure 17a), the multiplicative factor, that has essentially a pure annual signature, should disappear, or at least be attenuated. This is what is observed by comparing Figure 10d (the PW differences) and Figure 17b (the PW ratio) with respect to the other harmonic components. This line of reasoning is also valid for the wind that also has an annual signature, but with a greater variability and therefore a wider Fourier spectrum. It should be also noted that the mean slope between the THTI and FAA1 stations is −0.0344, a pretty close value, but this could be just a coincidence. Another argument is that Figure 16 clearly shows a causal time shift of about two months between the PW and insolation time series. This shift is certainly due to the response of the soil and/or vegetation to insolation [64,65,66]. Although these arguments are not evidence, we think that they point to the main reason, i.e., the difference in insolation between the THTI and FAA1 stations, to explain the differences between PW time series.

## 7. Conclusions

In this paper, we compared the PW estimates from two nearby permanent GPS stations (THTI and FAA1) in the tropical Tahiti Island (South Pacific) over a three-year period. The altitude difference between the two stations is only 86.14 m, and their horizontal distance difference is 2.56 km. Although these two stations are very close, they are located in very different surroundings. One of the stations is close to the sea shore (FAA1), on a very large flat terrain. The other one is located along the small Outumaoroa stream (THTI) along the slopes of the volcano, at 98.49 m altitude. In order to obtain reliable PW differences between the two stations, we processed the PW estimates for each station with great care, with a metrological validation of each step, and site tailored ZWD-to-PW relationships.

This study shows first that the differences between the two time series of PW values between THTI and FAA1 are not random, that they essentially contain an annual signature and other minor Fourier components, down to a diurnal signature. Secondly, the differences in altitude play only a minor role. The wind, through the transport of water vapor and evapotranspiration, plays a second role, with certainly a highly complex interaction between topography and winds [21,67]. In our case, the difference in insolation is certainly the key player. Moreover, this study points to the fact that a dense network of GPS stations, down to the hundred meters scale, will give constraints about the interaction between topography, surface temperature, wind, and evapotranspiration. This has been already noted by Mori et al. (2007) [68]. Another important conclusion is that when PW time series from isolated sites are assimilated in global meteorology or climate models, the associated formal error given to these time series at the end of the GNSS processing chain should be increased in order to take into account these micro-scale variations. 

## Figures and Tables

**Figure 1 sensors-19-05578-f001:**
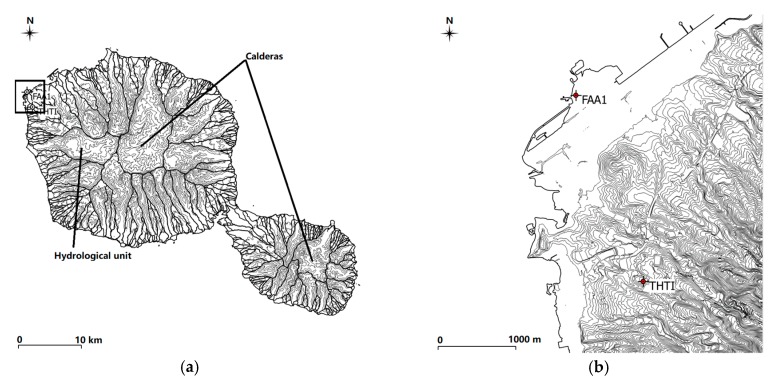
Location of the two nearby GPS stations (THTI (149.61° W, 17.58° S, ellipsoidal altitude 98.49 m) and FAA1 (149.62° W, 17.56° S, ellipsoidal altitude 12.35 m) in the Tahiti Island. (**a**) Tahiti Nui shield volcano (30 km in diameter) and Tahiti Iti volcano (15 km diameter), joined by the isthmus of Taravao. Tahiti is located in the South Pacific, at mid-distance from South America to Australia. The contour lines of the topography are every 200 m, with the limits of hydrographic units, mostly radial valleys, indicated by bold lines. (**b**) Enlargement of (**a**), near the two stations, with contour lines of the topography every 5 m. The airstrip is clearly visible on the enlargement, close to the FAA1 station. The calderas (volcano pits) are indicated in the figures as well as hydrological units (watersheds).

**Figure 2 sensors-19-05578-f002:**
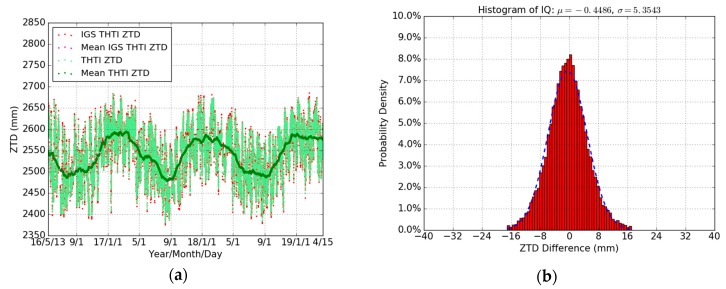
Comparisons of our THTI ZTD values (light-green dots) with IGS THTI ZTD (red dots), and monthly averaged estimates of our THTI ZTD (green dots) and IGS THTI ZTD (magenta dots) (**a**) and our FAA1 ZTD (light-green dots) with IGS FAA1 ZTD (red dots)), and monthly averaged estimates of our FAA1 ZTD (green dots) and IGS FAA1 ZTD (magenta dots) (**c**), and the histograms of IQ (intelligence quotient) for the corresponding ZTD differences between GPS THTI ZTD and IGS THTI ZTD (**b**) and between GPS FAA1 ZTD and IGS FAA1 ZTD (**d**).

**Figure 3 sensors-19-05578-f003:**
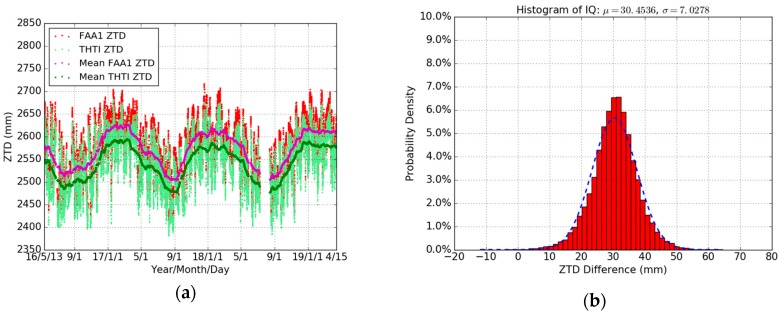
Comparisons between our THTI ZTD estimates (light-green dots) and our FAA1 ZTD estimates (red dots), and monthly averaged estimates of the THTI ZTD (green dots) and FAA1 ZTD (magenta dots) (**a**), and IGS THTI ZTD estimates (light-green dots) versus IGS FAA1 ZTD estimates (red dots), and monthly averaged estimates of the IGS THTI ZTD (green dots) and IGS FAA1 ZTD (magenta dots) (**c**), and the corresponding ZTD differences between THTI ZTDs and FAA1 ZTDs (**b**) and between IGS THTI ZTDs and IGS FAA1 ZTDs (**d**).

**Figure 4 sensors-19-05578-f004:**
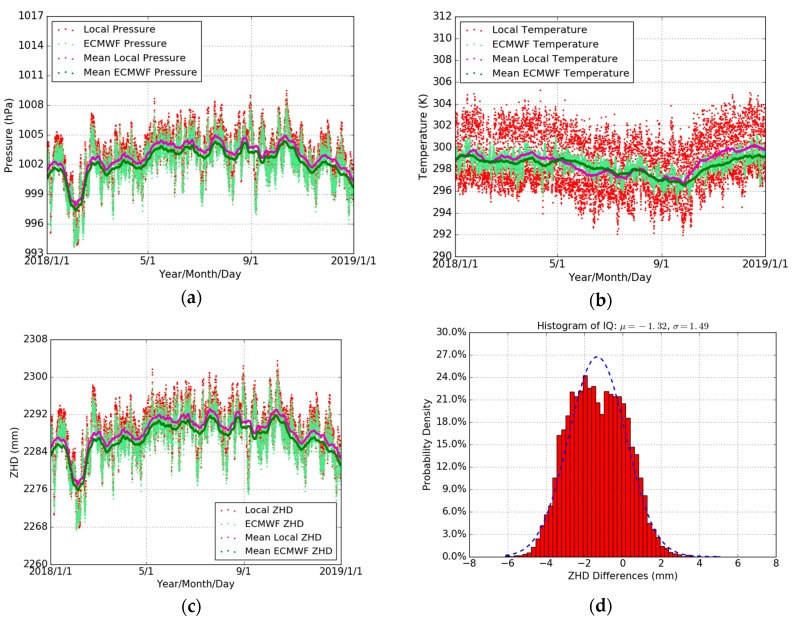
Comparisons of ECMWF pressure (light-green dots) with local pressure (red dots), and 10 days averaged estimates of the ECMWF pressure (green dots) and local pressure (magenta dots) (**a**), and ECMWF temperature (light-green dots) with local temperature (red dots), and 10 days averaged estimates of the ECMWF temperature (green dots) and local temperature (magenta dots) (**b**), and the time series of ZHD derived from ECMWF pressure (light green dots) and local pressure (red dots), and every 10 days averaged estimates of the ECMWF ZHD (green dots) and local ZHD (magenta dots) (**c**), and the histogram of ZHD difference (**d**) during the whole year of 2018. The temporal resolution is one hour.

**Figure 5 sensors-19-05578-f005:**
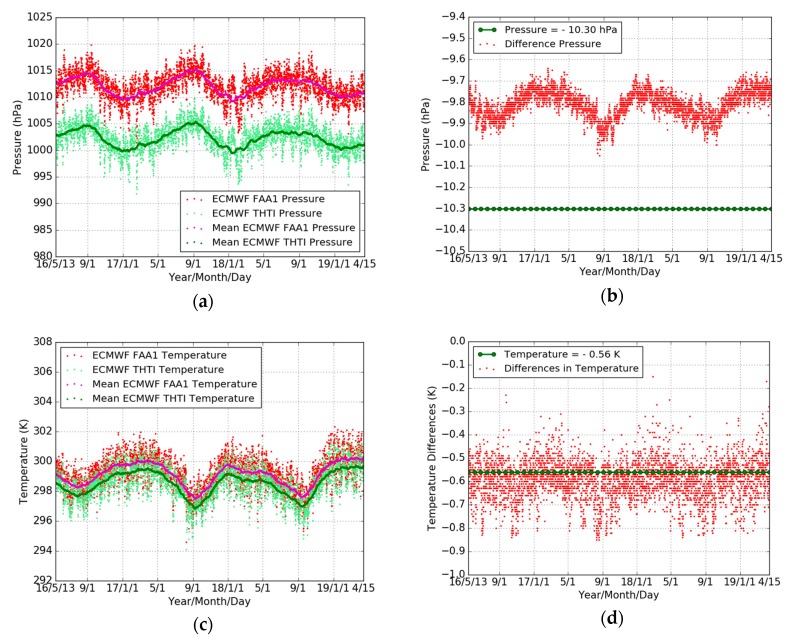
Comparisons of THTI (light-green dots) and FAA1 (red dots) ECMWF pressures, and monthly averaged estimates of the THTI (green dots) and FAA1 (magenta dots) ECMWF pressures (**a**) and temperatures (**c**), and the pressure difference from standard model (−10.30 hPa, green dots) and the corresponding pressure difference from ECMWF (**b**) the temperature difference from standard model (−0.56 K, green dots) and the corresponding temperature difference from ECMWF (**d**). The temporal resolution is six hours.

**Figure 6 sensors-19-05578-f006:**
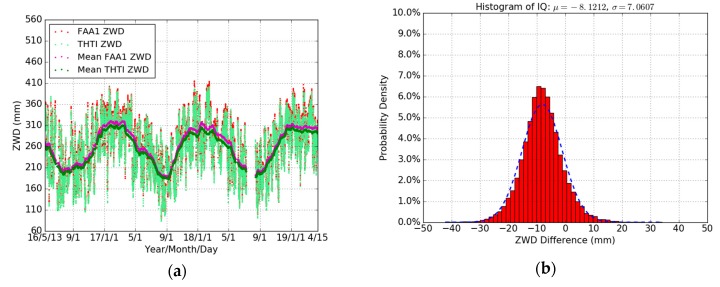
Comparisons of THTI ZWD (light-green dots) with FAA1 ZWD (red dots), and monthly averaged estimates of the THTI ZWD (green dots) and FAA1 ZWD (magenta dots) (**a**) and the ZWD difference between them (**b**). The temporal resolution is one hour.

**Figure 7 sensors-19-05578-f007:**
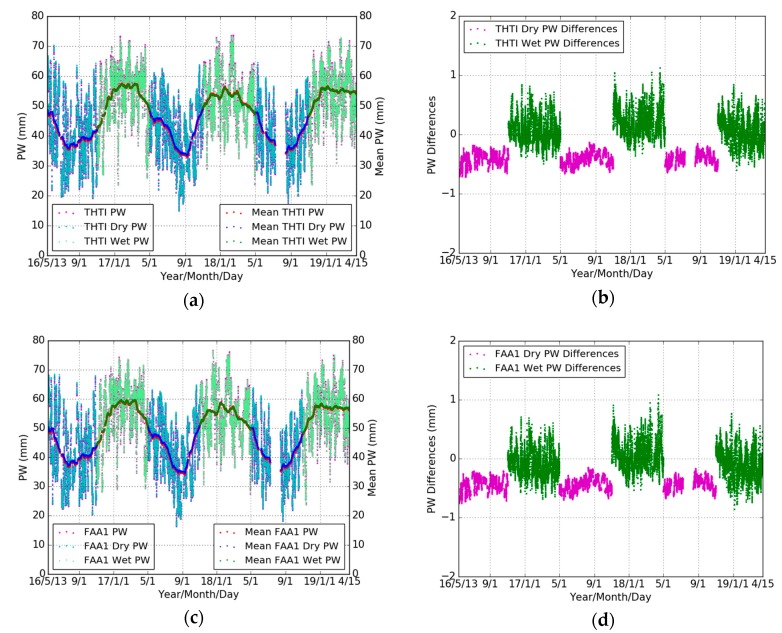
Comparisons of THTI PW values (magenta dots) based on all seasons’ $T_{m}$ model with THTI Dry PW (cyan dots) and THTI Wet PW (light-green dots) based on the corresponding dry and wet season’s $T_{m}\;$ models, and monthly averaged estimates of the THTI PW (red dots) and THTI Dry PW (blue dots) and THTI Wet PW (green dots) (**a**) and their differences (**b**), and comparisons of FAA1 PW (magenta dots) based on all seasons’ $T_{m}$ model with FAA1 Dry PW (cyan dots) and FAA1 Wet PW (light-green dots) based on the corresponding dry and wet season’s $T_{m}\;$ models, and monthly averaged estimates of the FAA1 PW (red dots) and FAA1 Dry PW (blue dots) and FAA1 Wet PW (green dots) (**c**) and their differences (**d**).

**Figure 8 sensors-19-05578-f008:**
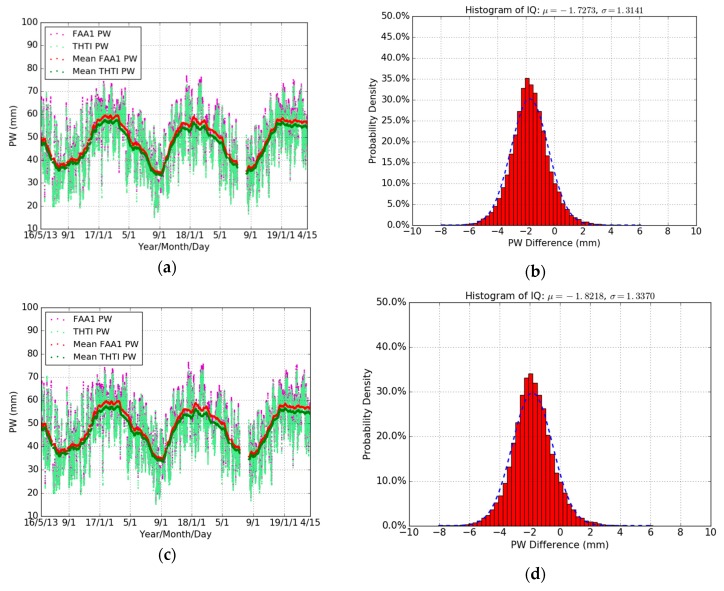
Comparisons of THTI PW (light-green dots) with FAA1 PW estimates (magenta dots) based on an all seasons’ $T_{m}$ model, and monthly averaged estimates of the THTI PW (green dots) and FAA1 PW (red dots) (**a**) and their differences (**b**), and based on seasonal $T_{m}$ models (**c**) and their differences (**d**).

**Figure 9 sensors-19-05578-f009:**
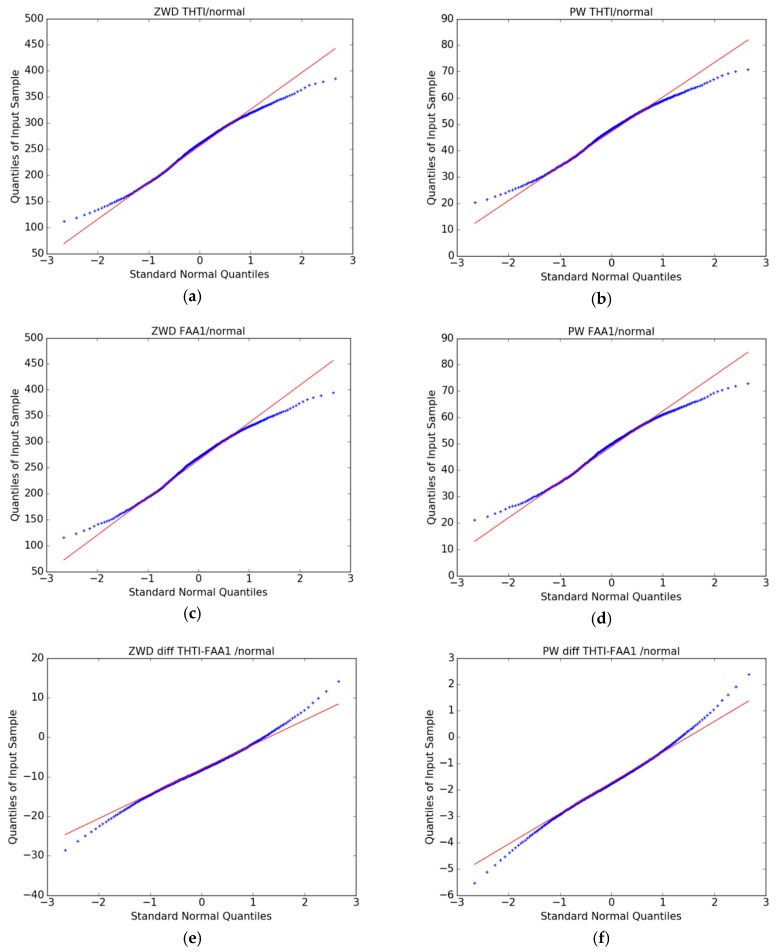
QQ plots of THTI ZWD (**a**) and THTI PW (**b**) with normal law, FAA1 ZWD (**c**) and FAA1 PW (**d**) with normal law, and QQ plots of cross-comparisons between THTI and FAA1, their ZWD difference with normal law (**e**) and their PW difference with normal law (**f**), and their ZWD difference with THTI ZWD (**g**) and their PW difference with THTI PW (**h**), and their ZWD differences with FAA1 ZWD (**i**), and their PW differences with FAA1 PW (**j**). The red lines are linear quartile–quartile estimates of the fit to be expected if the two distributions are linearly related.

**Figure 10 sensors-19-05578-f010:**
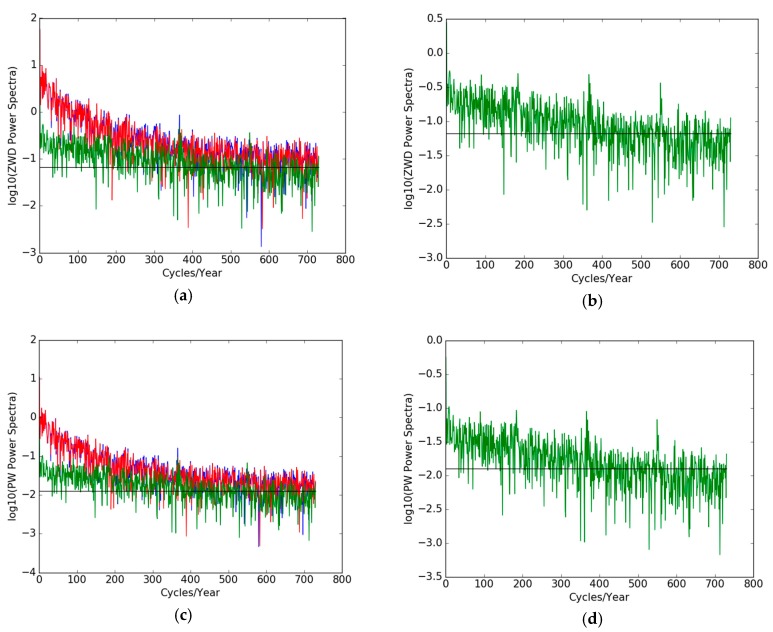
The power spectra of THTI ZWD (blue curve), FAA1 ZWD (red curve) and their differences (green curve) (**a**), and (**b**) the power spectrum of ZWD differences (enlarged green curve of (**a**)), the black line is the spectrum of a white noise matching the data noise; and the power spectra of THTI PW (blue curve), FAA1 PW (red curve) and their differences (green curve) (**c**), and (**d**) the power spectra of PW differences (enlarged green curve of (**c**)). The black line is the spectrum of white noise. Sub-diurnal variations cannot be retrieved, as they are buried in the noise.

**Figure 11 sensors-19-05578-f011:**
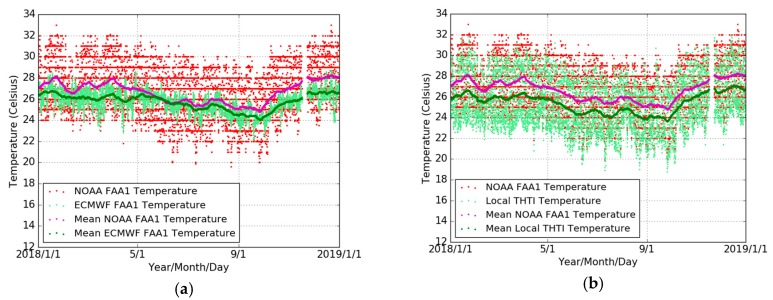
Comparisons of surface temperature from NOAA (red dots) and ECMWF (light-green dots), and 10 days averaged temperature estimates from NOAA (magenta dots) and ECMWF (green dots) (**a**), and the comparison of local THTI (light-green dots) temperature and NOAA FAA1 (red dots) temperature, and 10 days averaged temperature estimates of local THTI (green dots) and NOAA FAA1 (magenta dots) (**b**). The comparison is done for the whole year of 2018.

**Figure 12 sensors-19-05578-f012:**
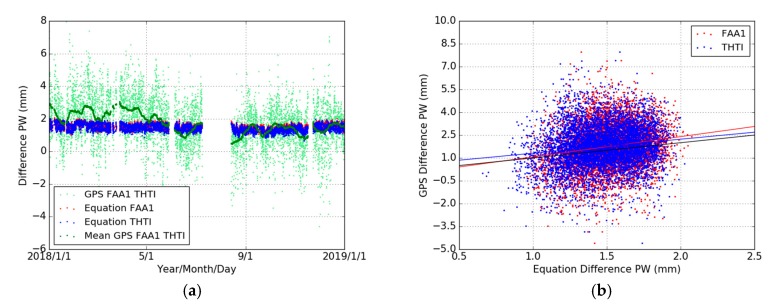
Comparisons of PW differences from GPS (FAA1-THTI, light-green dots) and their monthly averaged values (green dots) and exponentially derived PW (PWE) estimates (Equation (15)), with *n_s_* values for THTI (blue dots) and FAA1 (red dots) (**a**), and the respective fits of GPS-PW differences based on Equation (15) (**b**). The black line corresponds to the one-to-one relationship between the GPS-PW differences and the PWE differences.

**Figure 13 sensors-19-05578-f013:**
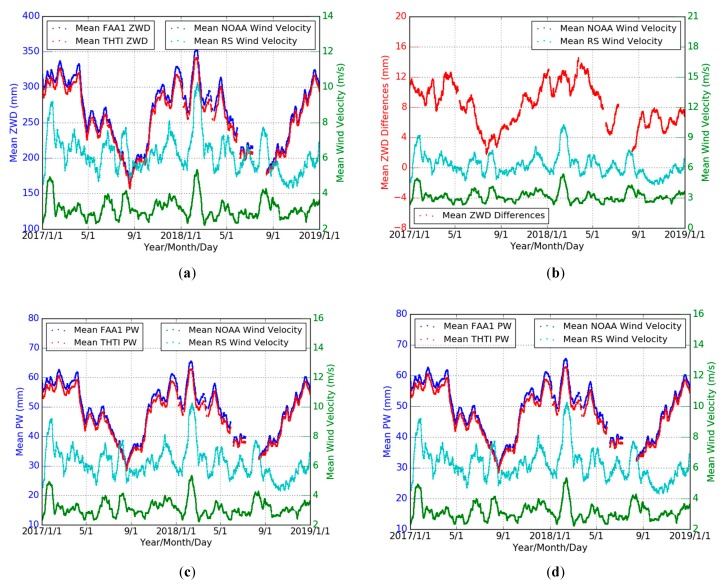
Variations of the averaged ZWD values of THTI (red dots) and FAA1 (blue dots) and the wind velocity values from NOAA (green dots) and RS (cyan dots) (**a**), and for PW values (**c**), and the variations of the corresponding averaged ZWD differences (red dots) between two stations with the wind velocity from NOAA (green dots) and RS (cyan dots) (**b**), and for PW differences (**d**).

**Figure 14 sensors-19-05578-f014:**
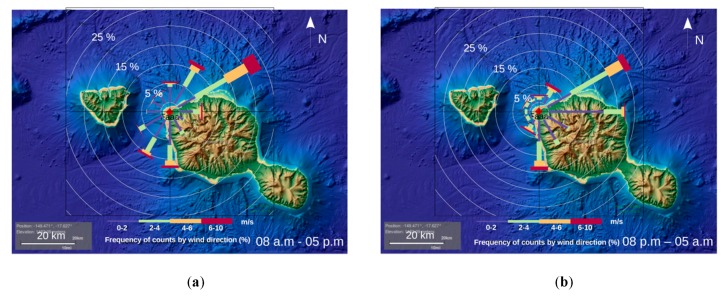
Ten m wind rose for 2017 and 2018 at FAA1 station: (**a**) during the day time from 8:00 to 17:00, and (**b**) during the night time from 20:00 to 05:00.

**Figure 15 sensors-19-05578-f015:**
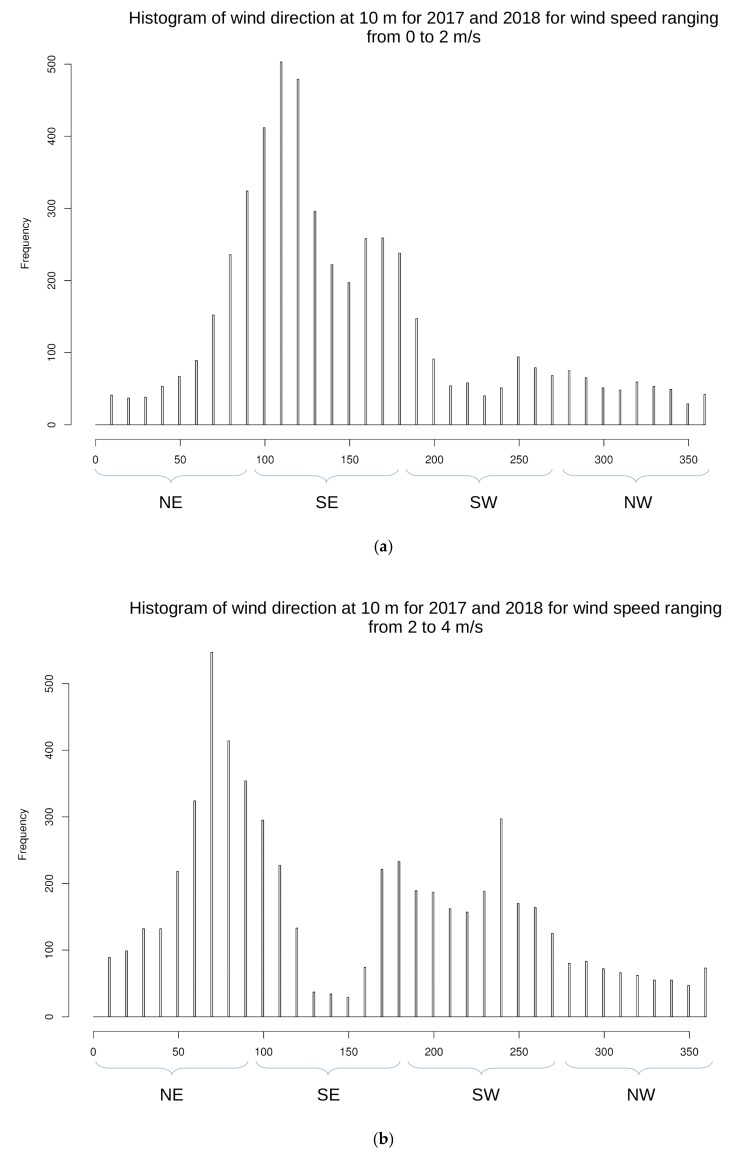
Histogram of 10 m wind direction at FAA1 for 2017–2018 (**a**) for 0 to 2 m/s, and (**b**) for 2 to 4 m/s, and (**c**) for 4 to 10 m/s. North-east direction is indicated by “NE” and spans 0° to 90°, south-east direction is indicated by “SE” and spans 90° to 180°, south-west direction is indicated by “SW” and spans 180° to 270°, north-west direction is indicated by “NW” and spans 270° to 360°. The corresponding wind rose is shown in Figure 14.

**Figure 16 sensors-19-05578-f016:**
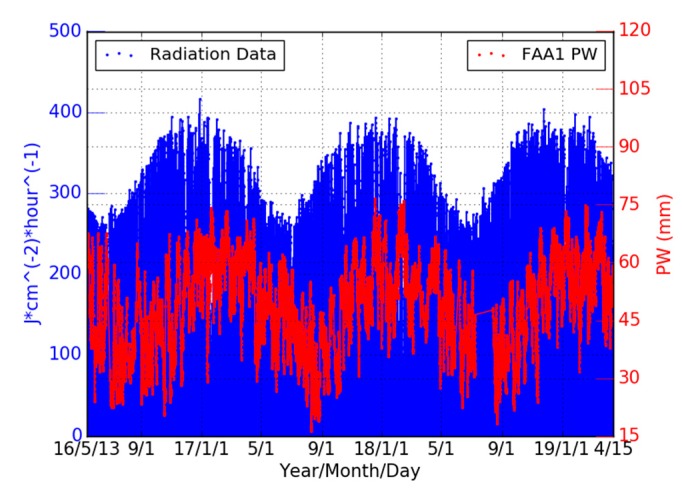
Hourly insolation variation from the pyranometer collocated with the FAA1 station, showing the strong annual signature driven by the Sun sky trajectory. The PW data relative to FAA1 station (see Section 6.7) are shown in red dots. A time shift of about two months between the two series is clearly visible, probably linked to the thermal inertia of the soil or/and a time delay in the vegetation response (evapotranspiration) to the insolation.

**Figure 17 sensors-19-05578-f017:**
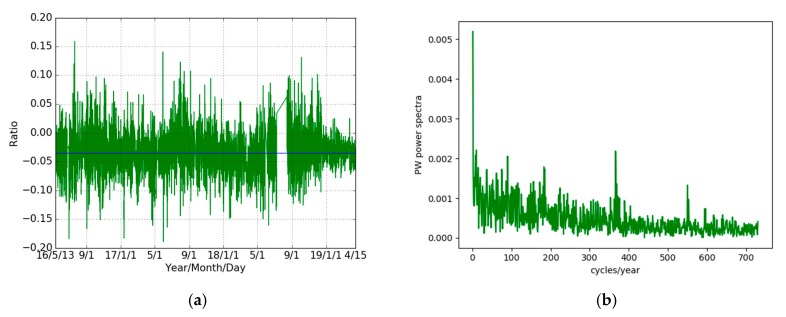
(**a**) Time series, over the three years of this study, of the ratio of PW values, defined as (PW(THTI)−PW(FAA1))/PW(FAA1). The mean value of this ratio is −0.0355 ± 0.0002. (**b**) Fourier analysis of the time series of subfigure (**a**). The annual component is largely attenuated with respect to the annual component in the time series of (PW(THTI)−PW(FAA1)), as seen in Figure 10d, pointing to a common factor with annual periodicity in the original time series of PW values for the two sites.

**Table 1 sensors-19-05578-t001:** Characteristics of our Global Positioning System (GPS) data processing and International Global Navigation Satellite System Service (IGS) data analysis centers.

	Our GPS Data Processing	IGS Products
Ephemerides, satellite clocks	CODE final	IGS final
Approach	PPP	PPP
Sampling interval	300 s	300 s
Elevation cut-off angle	3 degrees	7 degrees
Mapping function	Wet Vienna mapping function 1 (VMF1)	Wet global mapping function (GMF)
A priori zenith tropospheric delay (ZTD) model	European Center for Medium-Range Weather Forecasts (ECMWF)-based dry VMF1	Global pressure model (GPT)
Ocean loading	Applied	Applied
Atmospheric loading	Applied	Applied
Observables	Zero differences	Double differences
ZTD estimation interval	1 hour	5 minutes
Software version	Bernese 5.2	Bernese 5.0/Bernese 5.2

**Table 2 sensors-19-05578-t002:** $T_{m}$ models for all-season and two different tropical seasons (dry and wet) [20].

$T_{m}\;\mathrm{Model}$	Intercept	Sigma	Slope	Sigma
All seasons	−476.4	145.6	2.68	0.49
Dry season	−535.7	252.9	2.89	0.85
Wet season	−979.7	226.9	4.36	0.76

**Table 3 sensors-19-05578-t003:** Statistical summary for the comparison between GPS THTI/FAA1 ZTD and IGS THTI/FAA1 ZTD (IGS minus GPS) in terms of maximum (max), minimum (min), bias, root mean square (RMS) and standard deviation (STD), relative to Figure 2b,d.

Difference	Max (mm)	Min (mm)	Bias (mm)	RMS (mm)	STD (mm)	Data Points
THTI	17.09	−17.06	−0.45	5.37	5.35	24279
FAA1	26.76	−26.75	0.87	8.21	8.16	23741

**Table 4 sensors-19-05578-t004:** Statistical summary for the comparison between GPS/IGS THT ZTD and GPS/IGS FAA1 ZTD (FAA1 minus THTI) in terms of max, min, bias, RMS, and STD, relative to Figure 3b,d.

Difference	Max (mm)	Min (mm)	Bias (mm)	RMS (mm)	STD (mm)	Data Points
Our FAA1-THTI	64.53	−11.84	30.45	31.25	7.03	22679
IGS FAA1-THTI	71.90	1.20	31.71	32.17	5.42	22679

**Table 5 sensors-19-05578-t005:** Statistical summary for the comparison between ECMWF pressure/temperature and local pressure/temperature, and ZHD derived from two kinds of pressure data (ECMWF minus local) in terms of max, min, bias, RMS, and STD during the whole year of 2018.

Difference	Max	Min	Bias	RMS	STD	Data Points
Pressure (hPa)	2.68	−2.23	0.58	0.87	0.65	8726
Temperature (K)	6.33	−5.43	0.26	2.21	2.19	8726
DZHD (mm)	5.08	−6.12	−1.32	1.99	1.49	8726

**Table 6 sensors-19-05578-t006:** Statistical summary for the comparison between THTI and FAA1 ECMWF pressure/temperature (THTI minus FAA1) in terms of max, min, bias, RMS, and STD, relative to Figure 5b,d.

Difference	Max	Min	Bias	RMS	STD	Data Points
Pressure (hPa)	−9.64	−10.05	−9.81	9.81	0.06	4264
Temperature (K)	−0.15	−0.85	−0.60	0.60	0.09	4264

**Table 7 sensors-19-05578-t007:** Statistical summary for the comparison between our THTI ZWDs and our FAA1 ZWDs (THTI minus FAA1) in terms of max, min, bias, RMS, and STD, relative to Figure 6b.

Difference	Max (mm)	Min (mm)	Bias (mm)	RMS (mm)	STD (mm)	Data Points
ZWD	34.10	−42.22	−8.12	10.76	7.06	22649

**Table 8 sensors-19-05578-t008:** Statistical summary for the comparisons between THTI/FAA1 PW based on all seasons’ $T_{m}$ model and THTI/FAA1 PW based on seasonal $T_{m}\;$ model (all seasons’ minus seasonal) in terms of max, min, bias, RMS, and STD, relative to Figure 7b,d.

Difference	Max (mm)	Min (mm)	Bias (mm)	RMS (mm)	STD (mm)	Data Points
THTI Dry	−0.13	−0.72	−0.41	0.42	0.11	10,875
THTI Wet	1.13	−0.60	0.13	0.27	0.24	11,774
FAA1 Dry	−0.15	−0.76	−0.43	0.45	0.11	10,875
FAA1 Wet	1.09	−0.86	−0.03	0.25	0.25	11,774

**Table 9 sensors-19-05578-t009:** Statistical summary for the comparison between THTI and FAA1 PW estimates based on an all seasons’ $T_{m}$ model and based on seasonal $T_{m}$ models (THTI minus FAA1) in terms of max, min, bias, RMS, and STD, relative to Figure 8b,d.

Difference	Max (mm)	Min (mm)	Bias (mm)	RMS (mm)	STD (mm)	Data Points
All Seasons	6.11	−7.97	−1.73	2.17	1.31	22649
Seasonal	6.16	−8.05	−1.82	2.26	1.34	22649

**Table 10 sensors-19-05578-t010:** Statistical summary for the comparison between NOAA and ECMWF at FAA1 station (NOAA minus ECMWF) and the comparison of local THTI temperature with NOAA FAA1 temperature (NOAA FAA1 minus local THTI) in terms of max, min, bias, RMS, and STD, for the whole year of 2018.

Difference	Max	Min	Bias	RMS	STD	Data Points
FAA1 (°C)	6.87	−5.42	0.87	2.34	2.17	8271
FAA1–THTI (°C)	6.60	−2.00	1.23	1.44	0.76	8271

**Table 11 sensors-19-05578-t011:** Statistical summary for the differences of GPS-PW, PWE (FAA1) and PWE (THTI) in terms of max, min, bias, RMS, and STD, relative to Figure 12a.

Difference	Max (mm)	Min (mm)	Bias (mm)	RMS (mm)	STD (mm)	Data Points
GPS-PW	8.0	−4.59	1.75	2.24	1.39	6799
PWE (FAA1)	2.07	0.90	1.51	1.53	0.19	6799
PWE (THTI)	2.03	0.67	1.47	1.49	0.21	6799

**Table 12 sensors-19-05578-t012:** The fits of the difference of GPS-PWs and PWEs, relative to Figure 12b.

	Intercept	Slope	Correlation Coefficient
FAA1 PWEs	−0.26 ± 0.13	1.33 ± 0.09	0.18
THTI PWEs	0.39 ± 0.11	0.92 ± 0.08	0.14

**Table 13 sensors-19-05578-t013:** The fits of PW and ZWD differences between THTI and FAA1 with wind velocity from NOAA and RS.

	NOAA		RS		Correlation Coefficient	
	Intercept	Slope	Intercept	Slope	NOAA	RS
PW Difference	1.17 ± 0.06	0.18 ± 0.02	0.43 ± 0.07	0.21 ± 0.01	7.55%	16.24%
Mean PW Difference	1.24 ± 0.03	0.16 ± 0.01	0.47 ± 0.03	0.21 ± 0.00	16.25%	39.20%
ZWD Difference	5.58 ± 0.34	0.85 ± 0.11	1.67 ± 0.36	1.05 ± 0.06	6.43%	15.03%
Mean ZWD Difference	5.93 ± 0.13	0.73 ± 0.04	1.85 ± 0.13	1.02 ± 0.02	14.43%	37.94%

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
