# Peer review of "Analysis and Comparison of GPS Precipitable Water Estimates between Two Nearby Stations on Tahiti Island"

_sensors, 2019, doi:10.3390/s19245578_

Round 1

Reviewer 1 Report

Overall remarks:

The paper analyzed and compared the GPS precipitable water estimates between two nearby stations in Tahiti island. To obtain reliable PW differences between the two stations, great care was taken for the whole data processing, e.g., a detailed comparison with IGS products, a meteorological validation of each step, and site tailored ZWD-PW relationships. Some useful conclusions are given that the PW differences between two nearby stations is mainly driven by insolation differences, the difference in altitude and the wind being only second factors.

In general, the manuscript is well organized and written. It is well worth to communicate with readers. However, there are some questions which need to be addressed before accepting for publication, especially the confused data points in each table, which is pointed out below.

Specific comments:

The English language of the manuscript should be further checked, I gave an example here below, please go through the entire manuscript.

Line 56 “The nearest IGS station is located is on the Mangareva Island”.

I wonder if only Figure 2b, 2d removed off the outliers, while Figure 2a and 2c do not. It is observed that there is no data depicted between June 2018 and July 2018 in Figure 2c, please give the detail about this, and the same cases for Figure 3a and 3c.

Table 3 listed the statistical summary for the comparison between GPS THTI/FAA1 ZTD and IGS THTI/FAA1 ZTD, it indicated the accuracy of ZTD achieved based on your GPS data processing in THTI is slightly better than that in FAA1. Please explain it in more detail.

Reader may consider the data points in Table 4 are the results after removing off the outliers. Please give more information to explain that the data points are the same for the two cases in Table 4.

Figure 4 showed the comparison in a 1-min interval. Why the data points in Table 5 is only 8726 for a whole year. The same case for Table 6.

Line 283 Considering the ZHD with a 1-min interval and ZTD with a 1-h interval, please clarify the interval of ZWD estimates and the data points in Table 7.

Please add the legends for Figure 7a and 7c.

Author Response

Dear reviewer:

Thank you very much for your suggestions. 

We have completed the revision of this manuscript.  Attached is my response to your comments in detail. Thank you very much.

Best regards.

Fangzhao Zhang

Reviewer 2 Report

This paper analyzed the differences of PW between two nearby GPS stations (THTI and FAA1) in Tahiti based on a series of analysis including the statistical analysis by using QQ plots and a Fourier analysis, and the effect of the insolation, the altitude, and the wind velocities. Authors concluded that the differences in PW are mainly driven by insolation differences, the difference in altitude and the wind being only second factors. It is important to study the differences in PW time series, in order to assess the errors that can be induced in global models by the data from isolated stations, like Tahiti GPS stations. Therefore, this paper has enough quality to be published.

Author Response

Dear reviewer:

Thank you very much for your comments. 

Best regards.

Fangzhao Zhang

Author Response

(The authors gave the same response as above.)

Round 2

Reviewer 3 Report

I recommend the publication of the manuscript after some minor modifications without any re-review.